# MonoForest framework for tree ensemble analysis

**Igor Kuralenok**
Yandex / JetBrains Research
`solar@yandex-team.ru`

**Vasily Ershov**
Yandex
`noxoomo@yandex-team.ru`

**Igor Labutin**
Yandex / SPb HSE
Labutin.IgorL@gmail.com

## Abstract

In this work, we introduce a new decision tree ensemble representation framework: instead of using a graph model we transform each tree into a well-known polynomial form. We apply the new representation to three tasks: theoretical analysis, model reduction, and interpretation. The polynomial form of a tree ensemble allows a straightforward interpretation of the original model. In our experiments, it shows comparable results with state-of-the-art interpretation techniques. Another application of the framework is the ensemble-wise pruning: we can drop monomials from the polynomial, based on train data statistics. This way we reduce the model size up to 3 times without loss of its quality. It is possible to show the equivalence of tree shape classes that share the same polynomial. This fact gives us the ability to train a model in one tree's shape and exploit it in another, which is easier for computation or interpretation. We formulate a problem statement for optimal tree ensemble translation from one form to another and build a greedy solution to this problem.

## 1 Introduction

Industry and science combined efforts in the field of machine learning give us powerful techniques and tools to solve different kinds of supervised learning tasks. Just a few lines of code could train a model that solves classification, regression, ranking, and other problems. Modern techniques, like deep neural networks He et al. (11) learn complex models, but for many practical applications we need to understand why and how the model makes a prediction. This knowledge allows us to improve quality, protect from adversarial attacks, make a model resistant to data corruptions and so on. Recent efforts in deep learning models interpretation allow us to understand the decision for particular examples Ribeiro et al. (26); Shrikumar et al. (27); Štrumbelj, Kononenko (31); Koh, Liang (15); Lundberg, Lee (19), but understanding the logic behind a complex model is still a challenging task.

Unlike neural networks, decision trees are supposed to be easy to understand, which is true in case of shallow trees, but becomes a complicated task if we start using ensembles or increase the depth of a tree. Ensemble methods, especially gradient boosted decision trees, show state-of-the-art results on structured and categorical data Prokhorenkova et al. (25). For well-engineered input features decision tree ensembles significantly outperform deep networks: in competitions held by Kaggle ensemble models built by GBDT libraries often outperform their rivals.

One way to work with the interpretation problem is to make a clearer representation of a model or an optimization setup. An example of such perspective change is (4): the results of this paper allow us to use known facts and intuitions from differential equations for neural networks analysis.

Our work proposes a new framework for tree ensemble representation, learned by techniques like Random Forest (2) or Gradient Boosted Decision Tree Friedman (9). We call it MonoForest as sum of trees is converted to forest of monomials. The proposed framework can be used for both theoretical

tree ensemble analysis and practical enhancements of existing methods. The main contributions of this paper are:

- an algorithm for tree ensemble conversion to polynomial form;
- a proof of uniqueness of the polynomial form for all shapes of trees that have the same values;
- an algorithm for conversion of polynomial ensemble representation to sum of symmetric oblivious trees;
- an ensemble-wise pruning algorithm and its experimental study;
- a method for global feature attribution, based on polynomial form of tree ensemble.

The rest of the paper is organized as follows: section 3 introduces the new framework; the next three sections study applications of the proposed framework: theoretical analysis is in section 4, model reduction is in section 5, and interpretation is in section 6. Finally, section 7 summarizes future work and presents our conclusions.

## 2   Notation

At first, let us introduce a notation that will be used throughout the paper. $I\{*\}$ denotes the indicator function, i.e., $I\{\text{condition}\}$ is equal to one if the condition is true and zero otherwise. The vector of input features will be denoted by $x \in \mathbb{R}^n$. We assume that the decision tree ensemble is fixed, therefore there are a fixed number of minimal possible split rules $C$. For the rules we will use the following notations: $c(x) \in \{0, 1\}$ as a right split indicator function, i.e. $c(x) = I\{x_{i(c)} > b(c)\}$, where $i(c)$ is an index of feature to split, and $b(c)$ is a condition used; associated left split indicator, $I\{x_{i(c)} \leq b(c)\}$, is equal to $1 - c(x)$. We can group conditions by feature: $C_k = \{c \in C : i(c) = k\}$ and order them by condition value: $c_{ki}, c_{kj} \in C_k, b(c_{ki}) < b(c_{kj} \leq b_k) \Leftrightarrow i < j$ with maximal condition $b_k$ for each feature. $H$ is used for the entire ensemble, and $h_t$ for the components of the ensemble, e.g. $H(x) = \sum_{t=1}^{T} h_t(x)$; for the fixed decision tree we use $d$ to denote depth of the tree. By $2^C$ we denote the set of all possible products of right split indicator functions; elements of this set we call monomials and denote by $M$.

Popular boosting libraries have different growing policies for decision tree induction. All these policies use a greedy algorithm to search for a tree, but in a slightly different manner. As a result, libraries generate trees of different shape:

XGBoost Chen, Guestrin (5), by default, use a level-wise policy. Trees are built level by level, on each iteration each leaf is independently split by condition with the best gain. Optionally, the tree is prunned after training. This policy generates ensemble with balanced trees of fixed depth.

LightGBM Ke et al. (13) uses a loss-guide policy. This policy on each step of tree induction splits only a single leaf, one with the best gain. As a result, trees are usually highly imbalanced and deep.

CatBoost (25) uses a level-wise policy, similar to one used in XGBoost, but with one more restriction: CatBoost searches for just one single condition to split all leaves simultaneously. Thus, the result is, effectively, a decision table. Such trees are also called oblivious trees (16).

Throughout this paper we will refer to trees, generated by XGBoost and LightGBM as non-symmetric trees, and to trees, generated by CatBoost, as symmetric.

## 3   Tree ensemble as polynomial

The core idea of MonoForest is quite simple: we transfer a single tree to algebraic form and then minimize the entire ensemble, using properties of a tree representation. Decision tree splits the features space into non-intersecting areas (leaves $l$) where a single constant or vector is used to specify leaf property. It is possible to express this statement in algebraic form:

$$h(x) = \sum_{l \in \text{leaves}} w_l I\{x \in l\} \tag{1}$$

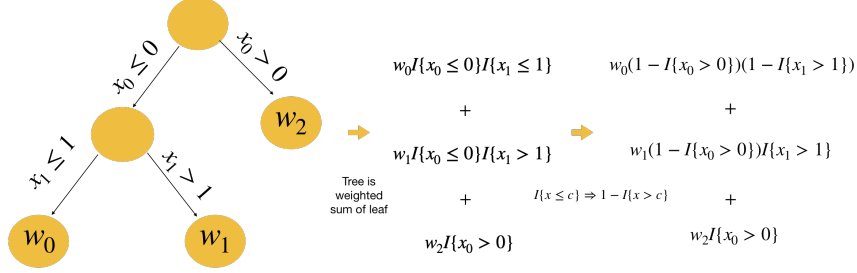

Figure 1: Core idea of MonoForest framework

Each leaf indicator function is a product of indicators induced by splits along the path from root to terminal node:

$$I\{x \in l\} = \prod_{c \in \text{right splits}} c(x) \prod_{c \in \text{left splits}} (1 - c(x)) \qquad (2)$$

Thus, replacing leaf indicator from equation (1) with equation (2) and expanding brackets, we will obtain tree in polynomial form:

$$h_t(x) = \sum_{M \in 2^C} w_{M_t} \prod_{c \in M} c(x) \qquad (3)$$

Figure 1 illustrates the idea of the transformation. The atomic parts of the representation $(w_M \prod_{c \in M} c(x))$ we refer to as monomials through the rest of this paper. The proposed representation of the tree has three valuable properties:

1. the values $w_{M_t}$ are defined by points, satisfying *all* conditions of the monomial;
2. conditions in a single monomial are based on different features;
3. it is possible to reorder conditions inside monomials.

The first property seems to be obvious, but it has interesting consequences: the set of points that define the value $w_M$ reduces when the number of conditions increases and the value itself becomes noisy; for all trees in the ensemble values $w_{M_t}$ are dependant on the exact same set of points and consequently depend on each other. The level of dependency grows with the number of conditions in $M$[1]. In extreme case, conditions from $M$ can split a point from the others and all $w_{M_t}$ are defined by this single point.

The second property comes from the fact that stronger conditions devours weaker ones: if two conditions of the same monomial are based on the same feature, their borders are always ordered and we can remove the weakest from the monomial. For example, assume leaf is a product of two indicator functions $I\{x_0 > 0\}I\{x_0 \le 1\}$. This is equal to $I\{x_0 > 0\}(1 - I\{x_0 > 1\}) = I\{x_0 > 0\} - I\{x_0 > 0\}I\{x_0 > 1\} = I\{x_0 > 0\} - I\{x_0 > 1\}$. $I\{x_0 > 1\}$ is a stronger condition in the second monomial, thus $I\{x_0 > 0\}$ could be removed. This property limits the number of possible monomials to $\sum_{i=1}^{d} \binom{n}{i}(\max_k |C_k|)^i$, which is significantly smaller then the naive estimate $\sum_{i=1}^{d} \binom{|C|}{i}$. Please note, that despite this fact, for simplification of notation we are using $2^C$ as set of possible monomials.

The third property allows optimizing model application time. For example, if there are two conditions that are met in the result model together and one condition $c_1$ is much more computationally difficult (e.g. taken from a database) than the other $c_2$, we can skip its calculation completely for examples $x$ such that $c_1(x) = 0$.

Summation over elements of ensemble gives us the final form of a decision function:

$$H(x) = \sum_{M \in 2^C} \left( \sum_{t=1}^{T} w_{M_t} \right) \prod_{c \in M} c(x) = \sum_{M \in 2^C} w_M \prod_{c \in M} c(x) \qquad (4)$$

This transformation alone is able to reduce the number of monomials comparing the leaves count in the original model due to the grouping of monomials from different trees.

To the best of our knowledge, this form was not studied before in the literature. The polynomial form is easier to analyze, interpret and it provides a way to work with different tree shapes in an unified manner.

**Implementation and conversion complexity**  Polynomial conversion could be done in a straightforward recursive way: we could extract leaves from the tree one-by-one. Each leaf produces at most $2^d$ monomials, where $d$ is the depth of this leaf. This conversion could be done in $O(|L|2^d)$, where $|L|$ denotes the number of leaves in the decision tree and $d$ is maximum depth.

## 4   Theoretical analysis

Decision tree equivalence, normalization, and minimal representation are known problems in statistics Lavalle, Fishburn (17) and computer science Zantema, Bodlaender (30). The proposed framework gives one more way to view these problems: we made a decomposition of complex decision function onto atomic decision factors with a certain degree of features interaction. In this section we use this property to set up a task of tree shape change in ensemble. We represent a tree ensemble as a sum of trees of fixed shape and minimize the length of this sum. To be able to declare the existence of such transfer we need to define equivalence class of tree ensembles with the following theorem:

**Theorem 1.** *Two tree ensembles $H$ and $H'$ are defined on $\mathbb{R}^{n}$[2], have the same value for all possible points $\forall x \in \mathbb{R}^n, H(x) = H'(x)$, iff: 1) their set of conditions $C$ and $C'$ are equal, 2) they have equal polynomial representation $\forall M \in 2^C, w_M = w'_M$*

*Proof.*  The reverse part of the proof is obvious. To prove the direct first proposition suppose, without loss of generality, that there is a condition $c \in C, c \notin C'$. We can take $x', x'' \in X$ such that $H(x') \neq H(x''), x'_{i(c)} = b(c), x''_{i(c)} = b(c) + \epsilon$ and, $x'_j = x''_j, j \neq i(c)$ because $C$ is a minimal conditions set for ensemble $H$, otherwise there is no such pair of points in $X$ that are split by $c$. Thus $c$ can be excluded from $C$. We can find $\epsilon$ such that $C'$ won't be able to split $x'$ from $x''$ because it does not contains $c$, consequently $H'(x') = H'(x'')$. This is contradiction of the initial statement.

Conditions sets are equal and we need to show that $\forall M, w_M = w'_M$. It is clear that $w_\emptyset = w'_\emptyset$ and we need to prove the induction step by size of $M$ and going from lowest to highest condition values for each of feature in set. For each features combination $C_M = \{i(c)|c \in M\}$ we start from the lowest conditions set $M_i = M_0 : c \in M_0 \Rightarrow b(c) = c_{i(c)0}$ and proceed to highest $\bar{M} : c \in \bar{M} \Rightarrow b(c) = b_{i(c)}$ rising single condition at a time. For this monomial $M_i$ choose $x', x'' \in \mathbb{R}^n$ such that:

- for all coordinates outside $C_M$ they both have value of minimal bound for this feature: $x'_k = x''_k = c_{k0}$,

- for all coordinates from $c \in M_i$: $x'_k = x''_k = b(c) + \epsilon$

For these points the difference between values of ensembles comes only from monomials from features set $C_M$: $0 = H(x') - H(x'') - (H'(x') - H'(x'')) = \sum_{j=0}^{i} w_{M_j} - w'_{M_j}$, because all monomials of lesser length are supposed to be equal in $H$ and $H'$, monomial of length $q$ with higher conditions ($j > i$) are zero by construction of $x'$ and $x''$. At first step there is the only element left in the right part of the equation: $0 = w_{M_0} - w'_{M_0}$ and $w_{M_0} = w'_{M_0}$, constructing pair of $x$ at each step of induction it is easy to show, that all $w_{M_i} = w'_{M_i}$. □

The direct implication of the theorem is that ensembles of trees that are able to generate the same set of monomials (e.g. of the same depth) are equivalent. Theoretically, this allows us to claim equivalence of the decisions set used by libraries with different shapes of atomic trees. Polynomial form determines the tree ensemble, but it is possible to convert the polynomial form back to a tree ensemble of a different form. This task could be formalized in the following way:

$$\{h_t^*\}_{t=1}^T = \underset{T, h_t \in \mathbb{H}}{\arg\min} \, \mathbb{E}_{x \sim X} \left( \left( H(x) - \sum_{t=1}^T h_t(x) \right)^2 \right) + \lambda T \tag{5}$$

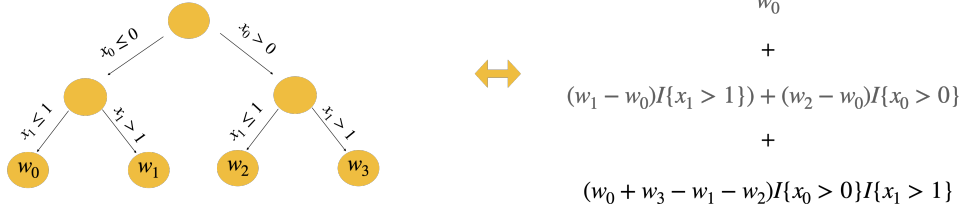

Figure 2: Oblivious tree conversion to polynomial form

where $\mathbb{H}$ is the set of trees of a certain shape and $\lambda$—regularization parameter. This set optimization is clearly NP-hard: there is a set cover problem under the shell[3]. On the other hand, the target of the optimization is submodular, so it is possible to find an approximately optimal solution in polynomial time Bach (1). Unfortunately, set optimization algorithms often become impractical because of their computational difficulty, and we decided to start from a simple greedy algorithm, leaving the mathematically sound version for future work. We have fixed the tree shape to symmetric oblivious trees. This type of trees is trivial in computations and it is possible to speed up decision function computation if we could convert tree ensembles of arbitrary shapes to them. From a practical perspective, the resolution of this task allows us to train easier for training but heavier model and then, transfer the result solution to some lightweight form, easier for exploitation.

The polynomial form of the ensemble gives the idea of a greedy algorithm for such optimization: at each step, we eliminate monomial with the greatest number of features in it. Using the fact that oblivious trees have a single monomial of maximum length (see Figure 2), the task becomes much easier. The resulting algorithm is presented in Algorithm 1.

There is no universal strategy for tree induction: optimal one changes with dataset at hand. On the other hand, symmetric trees outperform other tree shapes: they are coded by a decision table, thus evaluation for one tree requires several bit-wise operations and one look-up in the table and takes >10x less time for the same number of trees Dorogush et al. (6).

---

**Data:** Monomials $M_1 \ldots M_N$, monomial weights $W_1, \ldots W_N$
**Result:** Symmetric tree ensemble $H(x) = \sum_{t=1}^{T} H[t](x)$
**def** *SymmetricTree(M: monomial features, W : weight)*:
  | return oblivious tree, generated by monomial $M$ with weights $W$;
**def** *IsSubset(M : monomial features , h : tree)*:
  | return True if features from $M$ are subset of feature of tree $h$;
**def** *AddMonomialToTree(M : monomial features , W : weight, h : tree)*:
  | Add monomial $M$ with weight $W$ to tree $h$ ;
$H = []$;
$T = 0$;
**for** $i \in 1 \ldots N$ **do**
  | **if** $\exists t \in 0 \ldots T : IsSubset(M_i, H[t]) = True$ **then**
  | | /* Tree has the same split conditions                                    */
  | | AddMonomialToTree($M_i, W_i, H[t]$);
  | **else**
  | | $H[T] = \text{SymmetricTree}(M_i, W_i)$;
  | | $T = T + 1$;
**end**
Return $\sum_{t=1}^{T} H[t]$ ;

**Algorithm 1:** Greedy ensemble composition algorithm.

---

For our experiment we have used model built by LightGBM for Higgs dataset[4], transferred it to an ensemble of symmetric trees by Algorithm 1 and then compared model execution times of the original

model and transformed version. The time we need to apply the original model with fast CatBoost calculator for non-symmetric trees was 2.57 seconds; the transformed ensemble was applied in 1.57 seconds[5]. This gives us 40% speedup free of charge.

## 5   Model reduction

Decision tree pruning has been studied for decades: critical values, error complexity and reduced error methods (24), more recent bottom-up methods (14), minimum description length based methods (23) and others. Ensemble-wise pruning is less studied. Several works were done on the pruning of trained ensemble. (21), (22) trained a random forest ensemble, later using boosting to prune it. On each boosting iteration, tree search spaces were restricted to only those that were part of a random forest. Kappa pruning Margineantu, Dietterich (20), and modification of this technique (29), were proposed to prune AdaBoost ensembles. Those heuristic techniques rely on the assumption that gradient boosting builds an ensemble of weak classifiers. Greedy strategies are then used to select the most diverse sample of them. To the best of our knowledge, those methods are not used in practice today. In practice, pruning is done on a per-tree basis, and early stopping strategies are used to select the optimal ensemble size Chen, Guestrin (5); Prokhorenkova et al. (25).

In this part we use the first property of the polynomial representation: the monomial coefficients are determined by the same set of points in all trees of the ensemble. We will define a quality measure of the resulting monomials, based on point statistics and remove the least valuable from the model. To measure monomial quality we use $\eta(M) = \mathbb{E}_{x \sim X} \left( w_M^2 \prod_{c \in M} c(x) \right)$. This approach is closely related to feature importance, proposed by Breiman et al. (3). They estimated a squared risk improvement from region partitioning; our measure is an estimated squared risk improvement over setting the specified monomial weight to zero. This way we get a simple ensemble-wise pruning strategy while it is definitely possible to use more sophisticated methods like sparse re-weighting monomials.

The introduced monomial quality measure leads to a straightforward pruning scheme: select some threshold $\alpha$ and for each monomial $M$ with $\eta(M) < \alpha$ set weight to zero. This threshold could be selected by heuristics like 'Elbow method', based on learn statistics only, or $\alpha$ could be estimated using the validation set. The former approach requires human judgment; thus its quality is hard to estimate. The latter one could be applied automatically; thus we could estimate its performance in a fair way.

We use several publicly-available binary classification datasets[6] to evaluate the quality of the automatic approach. The experimental setup is the following: we split the data into train/validate/test groups, the parameters were tuned on train/validation pair, then these parameters were used on train+validate joined dataset to get the final model. The result model is evaluated on the test part of a data. It is important to note that we tuned the optimal gradient step, number of trees in the ensemble, regularization parameters of a single trees, etc. This way we get an ensemble with the minimal number of leaves using state-of-the-art techniques.

Obtained ROC AUC values are presented in the second (original model) and third (pruned model) columns of the Table 1. The last column contains the polynomial model reduction ratio. The experimental results allow us to claim that the tree ensemble can be significantly reduced without loss of model quality in a variety of practical tasks. The features used there have numerical as well as categorical nature.

## 6   Interpretation

Feature attribution methods are designed to answer, why and how each feature influences models predictions. These methods could be global, describing feature influence on average, or local, explaining how the model deals with one sample.

Global feature attribution methods are well developed. Classical approaches are still widely used today. Breiman et al. (3) gain, or total reduction of loss contributed by all splits, provides a way

| DataSet | Trained ensemble(AUC) | Pruned ensemble(AUC) | Model reduction (Ratio) |
|---|---|---|---|
| Adult | 92.76% | 92.75% | 2.58 |
| Amazon | 82.51% | 82.51% | 2.6 |
| KDD Internet | 95.71% | 95.74% | 2.07 |
| KDD Upselling | 85.72% | 85.72% | 3.78 |
| Epsilon | 95.76% | 95.76% | 1.11 |

Table 1: Quality and size of original and pruned models.

to estimate the relative contribution of each input feature to the response. Partial dependency plots Hastie et al. (10) are used to summarize dependence of response on the input feature. Resampling strategies are used to design alternatives to Breiman et al. (3) feature importance measures. Brieman's 'Variable Importance' Breiman (2) for random forests and its model-agnostic version 'Model Reliance' Fisher et al. (8) are the best known examples. Other sampling strategies lead to a big variety of others Díaz-Uriarte, Andrés de (7); Ishwaran (12); Strobl et al. (28).

Local feature attribution methods deal with importance measures for each sample in the dataset. Work by Lundberg, Lee (19) has recently shown that, under certain conditions, there is a single unique solution for additive feature attribution methods with three desirable properties (local accuracy, missingness, and consistency) — SHAP values. This work was adapted to decision trees in (18).

All these techniques could be applied combined with MonoForest framework. However, our representation allows us to interpret a tree ensemble model as a linear function. As shown in Section 3, it is possible to present a tree ensemble as polynomial:

$$H(x) = \sum_{M \in 2^C} w_M \prod_{c \in M} I\left\{ x_{i(c)} > b(c) \right\} \tag{6}$$

Due to the second property of the representation, for each feature $k$ monomial $M$ either contains a single condition $c$ such that $i(c) = k$, or there is no dependency on this feature. Let us denote $C_{-k} = \{c \in C : i(c) \neq k\}$, $c_{ki}$—$i$-th ordered border condition on feature $k$: $i(c) = k, i > j \Leftrightarrow b(c_{ki}) > b(c_{kj})$. We can redistribute monomials the following way:

$$H(x) = \sum_{M \in 2^{C-k}} w_M \prod_{c \in M} c(x) + \sum_i I\left\{ x_k > b_{ki} \right\} \left( \sum_{M \in 2^{C-k}} w_{M \cup \{c_{ki}\}} \prod_{c \in M} c(x) \right) \tag{7}$$

This formula is linearly dependant on conditions $I\left\{ x_k > b_{ki} \right\}$ and we can use expected linear coefficients to evaluate the influence of each condition $c_{ki}$. To get a single value for entire feature we sum all dependant condition values:

$$\nu(k) = \sum_i \mathbb{E}_{x \sim X} \left( I\left\{ x_k > b_{ki} \right\} \sum_{M \in 2^{C-k}} w_{M \cup \{c_{ki}\}} \prod_{c \in M} c(x) \right) \tag{8}$$

The aggregation to the final feature score is straightforward and has its limitations. For example if the monomial values are spread around zero throughout the data points it will have next to zero expectation despite the possible big influence of the feature to particular points. The naive nature of the proposed score does not interfere with its good behavior in real-world examples.

There is no established way to compare feature attribution methods. To demonstrate the quality of the proposed approach we built a decision tree ensemble binary one-vs-rest classifier for each class of the MNIST dataset using CatBoost and analyzed each classifier using three methods: MonoForest, SHAP, and permutation based Model Reliance, proposed by Fisher et al. (8). The model accuracy for each class was $> 98\%$. The task of the MNIST dataset is to identify a single handwritten digit. The features are the levels of grey of each pixel in the input image. We have calculated importance values for all features and put them on the picture of the same size as input images. SHAP is known as a local attribution method, but the authors suggested a way to use it as a global attribution too. The results for the first four classes are presented in the Figure 3. The MonoForest results allow splitting positive and negative influence of features in comparison to SHAP global attribution and Model Reliance.

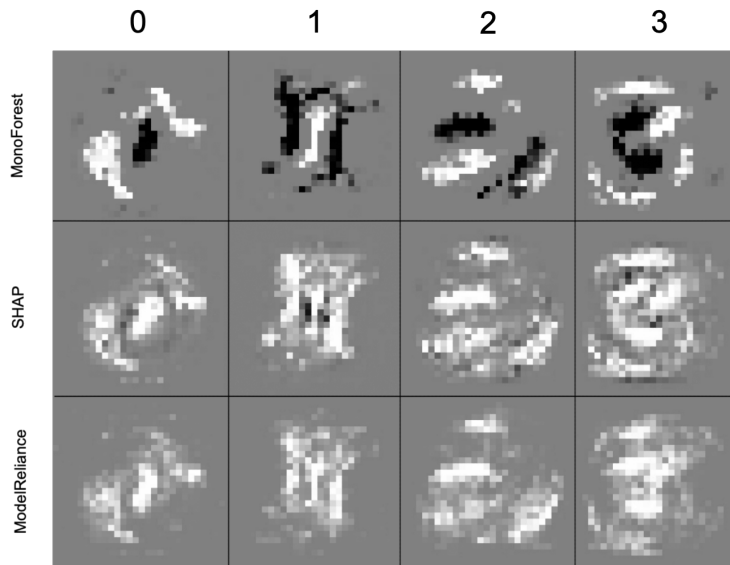

Figure 3: Feature importance visualisations on MNIST data. MonoForest, SHAP and Model Reliance results are presented in rows for digits 0, 1, 2 and 3.

# 7    Conclusions and Future work

In this work, we have introduced a new framework for decision tree ensemble representation. The proposed framework demonstrated good results on two popular tasks: model interpretation and reduction. The new representation has interesting mathematical properties of its components, and they allow us to simplify algorithms relevant for decision tree ensembles. We show that, using primitive filtering techniques for linear models, it is possible to significantly simplify the original model without loss of its quality after applying other pruning strategies.

Another important task for MonoForest application is model interpretation. We used a straightforward approach to polynomial models analysis and were able to get promising results in comparison with state-of-the-art techniques, such as SHAP and VI, though a comparison of interpretation methods is not well-established field yet.

The proposed representation allows us to introduce a class of equivalence in tree ensembles and claim that the decision space of the popular GBDT libraries is equivalent. We have set up a problem of optimal ensemble conversion and provided a greedy algorithm to solve this problem. It is important to note that this way we can split tasks of ensemble training and performance optimization of the result decision function. As an example of such conversion, we used easy-to-train LightGBM trees and then converted them to a more effective form of oblivious trees used by CatBoost. The algorithm can be improved by set optimization techniques to achieve even better results.

The MonoForest framework has shown its applicability to white-box analysis of decision trees ensembles. This work shows that even 'naive' methods provide good results, thus promising much more when combined with more sophisticated techniques. There are several directions for future explorations:

- framework applications expansion to other tasks;

- improvement of ensemble tree pruning algorithm: LASSO and other regularization based techniques look promising;

- greedy performance optimization of the model application: if we want to build ordered by model value list, it is possible to skip some computationally demanding features valuation

## Footnotes

[1]MonoForest decomposition is similar to $n$-way ANOVA decomposition, where dependence for factors $x, y, z$ is decomposed to main effects $(x, y, z)$ and their iterations $(xy, yz, xyz)$. In ANOVA $x, y, z$ are categorical factors, while in MonoForest it is right split indicator functions.

[2]To save the space we formulated the theorem in $\mathbb{R}^n$ task space, this limitation can be easily eliminated, but the proof will be longer.

[3]We need to cover all monomials by polynomials of the fixed shape.

[4]LightGBM provides the best quality model for this dataset

[5]For experiment we have used dual-socket server with Intel Xeon CPU E5-2650 and 256GB of RAM. CatBoost version was equal to 0.14.2

[6]Information about datasets is available in supplementary materials.

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
