[Supplementary Material]

| DataSet | Url | Numerical columns | Categorical columns | Sample count |
|---|---|---|---|---|
| Adult | https://archive.ics.uci.edu/ml/datasets/Adult | 6 | 8 | 48842 |
| Amazon | https://www.kaggle.com/bittlingmayer/amazonreviews | 0 | 9 | 32769 |
| KDD Internet | https://kdd.ics.uci.edu/databases/internet_usage/internet_usage.html | 47 | 21 | 10108 |
| KDD Upselling | http://www.kdd.org/kdd-cup/view/kdd-cup-2009/Data | 381 | 38 | 50000 |
| Epsilon | https://www.csie.ntu.edu.tw/~cjlin/libsvmtools/datasets/binary.html | 2000 | 0 | 500000 |

Table 1: DataSets.

# 1   Subroutines from greedy algorithm

Here we present c-style pseudo-code for routines, used in greedy algorithm.

```
struct RightSplitIndicator {
    float Border;
    int FeatureIndex;

    bool Value(const float* x) {
        if (x[FeatureIndex] > Border) {
            return true;
        }
        return false;
    }
}

struct SymmetricTree {
    int Depth;
    RightSplitIndicator Conditions[Depth];
    float Leaves[1 << Depth];

    float Value(const float* x) {
        int leaf = 0;
        for (int i = 0; i < Depth; ++i) {
            if (RightSplitIndicator[i].Value(x)) {
                leaf |= 1 << i;
            }
        }
        return Leaves[leaf];
    }
}

struct Monomial {
    int Depth;
    RightSplitIndicator Conditions[Depth];
    float Value;

    float Value(const float* x) {
        for (int i = 0; i < Depth; ++i) {
            if (Conditions[i].Value(x) == false) {
                return 0;
            }
        }
        return Value;
    }
}

SymmetricTree MonomialToSymmetricTree(Monomial monomial) {
    SymmetricTree tree;
    tree.Depth = monomial.Depth;
    tree.Conditions = monomial.Conditions;
    for (int i = 0; i < (1 << Depth); ++i) {
        tree.Leaves[i] = 0;
    }
```

```
        tree.Leaves[(1 << Depth) - 1] = monomial.Weight;
        return tree;
}

// is monomial subset of tree
// warning: this for clarity only, not the proper implementation
bool  IsSubset(Monomial monomial, SymmetricTree tree) {
        set<RightSplitIndicator> treeSplits(tree.Conditions);
        set<RightSplitIndicator> monomialSplits(monomial.Conditions);
        return monomialSplits.Subset(treeSplits);
}

// expects IsSubset(monomial, tree) == True
// warning: code is for algorithm clarity, not the real implementation
void AddMonomialToTree(Monomial monomial, SymmetricTree tree) {
        float* leaves = tree.Leaves();
        RightSplitIndicator* monomialSplits = monomial.Conditions;
        RightSplitIndicator*  otSplits = tree.Conditions;

        int treeDepth = tree.Depth;
        int monomialDepth = monomial.Depth;

        vector<int> bitsToFill;
        int baseLeaf = 0;
        {
                int monomialCursor = 0;
                int otCursor = 0;
                while (otCursor < treeDepth) {
                        if (polynomCursor < monomialDepth and
                            monomialSplits[monomialCursor] == otSplits[otCursor]) {
                                baseLeaf |= 1 << otCursor;
                                ++monomialCursor;
                        } else {
                                bitsToFill.push_back(otCursor);
                        }
                        ++otCursor;
                }
        }

        const int iterCount = 1 << bitsToFill.size();
        for (int i = 0; i < iterCount; ++i) {
                int leaf = baseLeaf;

                for (ui32 j = 0; j < bitsToFill.size(); ++j) {
                        if (i & (1 << j)) {
                                leaf |= 1 << bitsToFill[j];
                        }
                }
                leaves[leaf] += monomial.Value;
        }

}
```