[Reviews · NeurIPS 2019]

Reviewer 1



This paper contains several contributions. The authors propose a polynomial form of a tree ensemble. A tree is expressed as a sum (over leaf) of a weight (associated to the leaf) and a product of conditions. The authors provide some theoretical results linking trees and this representation. Based on this representation, they propose methods to compress tree ensembles, to prune them and to measure feature importance. They also provide some experimental results. Imho, this paper is rather original. While trees and ensembles have already been studied as a sum of weights over leafs, I do not think the methods derived by the authors have been considered before. Following the authors clarification, I think the paper is relatively clear and, with the modification the authors promised in the rebuttal, of a quality fit for publication. The authors make several contributions and back them with experimental results.I believe the details they make available about their experiments would make it reasonably easy to reproduce them. On the negative side, they do not compare their methods to baselines for the pruning experiment, they do not provide any code. In my opinion the paper would also benefit from additional proof-reading. My main concern was that the paper was often unclear. Here are the main points that bothered me initially, but the authored cleared them up in the rebuttal, promising to correct mistakes and pointed out some understanding mistake I made. I'm sorry for that. A) l74: I do not understand condition 2. A leaf can be bounded from the left and the right, so can't a feature be present twice? B) Theorem 1: Wouldn't it be possible to construct an ensemble H' from an ensemble H by splitting one leaf l and assigning the weigh of the old leaf in the new leaves in H'? Hence the ensemble would have the same value for all points but different set of conditions. C) equation 2 seems wrong with respect to right and left splits and the definition of c(x) equation 3: M does not seem to have been defined before. D) l 85: d does not seem to have been defined before. l105: In this section we (...) set up a task of tree shape change in ensemble. We represent a tree ensemble as a sum of trees of fixed shape. ---> unclear to me whether the shape is changed or not. Nevertheless, I believe the methods proposed to be interesting. The lack of comparison to existing approaches make it hard to assess the impact they can have on practitioners. However, the fact the authors derive various methods from the representation they study leads me to believe this impact could be high. some typos: 56: there are fixed number of rules 82: the stronger condition devours weaker 112: but proof will be longer 149: we took model --------------- I would like to thank the authors for pointing out misunderstandings on my side. The rebuttal was also very nice to understand the paper. I have updated my review above to take the rebuttal into account. I would like to stress again that I now think the paper to be worth publishing though there is still some room for improvement. I will update my score accordingly. I would encourage the authors to include the example included in l8-10 of the rebuttal in the paper. On a related note, I think there may be a mistake in these lines. Shouldn't the first equality on line 9 be "I(x > 0)(1 − I(x > 1)) = I( x >0 ) − I(x>0)I(x>1)" ? I also suggest improving the illustrative Figure 1 by including an additional step: the transformation of the sum into an expression containing only right split indicator functions. If the authors implement this and the changes they suggested in their answer, I think my complains would no longer be relevant.

Reviewer 2



This paper proposes using the polynomial representation for the trees. The main contribution of the authors can be summarized to the theoretical analysis, model pruning, and interpretation of the mentioned representation. In the theoretical analysis presented in the paper, the authors introduced a theorem to define the equivalence of the tree ensembles with the same set of conditions and polynomial representation. Also, an optimization problem is formalized for converting polynomial form back to tree ensemble. However, later for simplicity, a greedy algorithm is proposed to convert the polynomial form to the ensemble of symmetric oblivious trees. Based on PolyTree representation, a model reduction strategy is proposed in which the weight of the monomials with (defined) quality lower than a threshold is set to zero. Finally, an illustrative experiment and a theoretical justification are presented to show the ability of the presented model for feature importance. The main idea presented in this work, a novel polynomial representation form for the decision trees, is interesting. Mainly because it can open new avenues to understand and interpret the tree ensembles in general. The paper is easy to read and well organized. However, I have some concerns related to the experimental results presented in section 4.

Reviewer 3



In this paper, authors propose a new framework for tree ensemble representation while transforming each tree into a well-known polynomial form. The proposed repreentation is applied to three tasks namely theoretical analysis, model reduction, and interpretation. Experimentations are conducted ans some comparisons are made. The proposed idea is quite original by ensuring three tasks namely theorectical analysis, the model reduction and the interpretation. The contributions are somewhat interesting where authors propose an algorithm for tree ensemble conversion to polynomial form; a proof of uniqueness of the polynomial form for all shapes of trees, that have the same values is provided. Besides, author develop an algorithm for conversion of polynomial ensemble representation to sum of symmetric oblivious trees. They also deal with ensemble pruning, and also propose a method for global feature attribution. The results are in general sound, a proof of Theorem 1 is provided. Experiments are somewhat convincing but needed to be more detailed and more comparisons should be done. In general, the paper is well-organized with few typos to correct. However Figures 2 and 3 are not very clear and their titles are very long. Examples should be added make the article clearer. Even the paper presents somewhat interesting ideas, there are some comments that should be addressed : - The introduction needs more depth to better understand the motivation of the paper. - The paper lacks of examples which will better illustrate the proposed ideas. . - Why the level of dependency grows with the number of conditions in M. - For section 3 relative to theoretical analysis, experiments remain poor, testing on only one set (Higg dataset) is insufficient to make reliable conclusions. More data sets should be tested. - In Tabe 1, How to explain that there is no significant improvement In AUC criterion between trained ensembles and pruned ensembles. In fact, you get almost the same AUC for both methods - In Table 1, evaluating based only on AUC and model reduction is unsufficient other criteria should be added like the computational time, etc. - Authors do not mentionned if the cross validation is applied or not. This latter should be applied. - Why what is tested in Table 1 is only sets of binary classes. What about sets sets with multiple classes.

[Author Response · NeurIPS 2019]

We would like to thank all reviewers for their invaluable feedback. The next revision of the paper will include fixes for all typos that were mentioned. Responses for questions raised by each reviewer are below.

**Reviewer #1**

We would like to clarify questions about the theory you've raised. PolyTree framework represent tree and ensembles not as a sum of weights over **leaves**, but over **right split indicator functions** ($c(x) = I\{x > b\}$, see line 57). This fact is a core idea and should be kept in mind to understand most of the propositions: *A) l74: I do not understand condition 2. A leaf can be bounded from the left and the right, so can't a feature be present twice?* The feature will still be represented only once. Let look at leaf $x \in (0, 1]$. This leaf is product of two indicator function $I\{x > 0\}I\{x \leq 1\}$, this is equal to $I\{x > 0\}(1 - I\{x > 1\} = 1 - I\{x > 0\}I\{x > 1\} = 1 - I\{x > 1\}$. $I\{x > 1\}$ is a stronger condition, thus $I\{x > 0\}$ could be removed. We will add this example to the next revision of the paper to clarify this property. *Theorem 1: Wouldn't it be possible to construct an ensemble H' from an ensemble H by splitting one leaf l and assigning the weigh of the old leaf in the new leaves in H'?* When the ensemble is represented as a sum over leaves, there indeed will be two different ensembles. But, both ensembles will generate the same PolyTree representation, because we perform a reduction of weights for monomials with the same set of conditions (such leaf will generate 2 monomials with the same absolute weights, but different signs, after reduction terms will disappear). This comes from condition 2 (line 74). *l105: In this section, we (...) set up a task of tree shape change in an ensemble. We represent a tree ensemble as a sum of trees of fixed shape. —> unclear to me whether the shape is changed or not.* By fixed shape we mean a restriction on a set of possible tree structures. For example, tree of depth 6; balanced trees; symmetric oblivious trees and so on. *equation 2 seems wrong...* Thank you, this is a typo and will be fixed. *Comments about notations* By $2^C$ we denote set of all possible monomial structures, and $M \in 2^C$ represent some monomial from this set. We use letter $d$ to denote the depth of decision tree, and in the context of line 85 this is a maximum depth of tree in the ensemble. We'll add all this to paragraph with notation. *Provide code* We have Java-based implementation on GitHub and will include the link with non-anonymous version. We are also working on an implementation of proposed methods as a part of one of the major GBDT libraries and it'll be finished before NeurIPS2019.

**Reviewer #2**

*- In the last paragraph of section 3, the execution time of an original ensemble with the transformed one using the proposed algorithm is compared; nevertheless, the overhead time added by this transformation is not mentioned.* Our greedy algorithm is quite slow and could take several seconds to complete. Also, this experiment is a proof-of-concept and we did not optimize it to run as fast as possible. Mostly, such transformations are interesting for production tasks where the same model could be used for days. Transformation time could be big, but it is done once, thus overhead is negligible. *...Symmetric trees can highly benefit from model reduction because of having many zero leaves...* For more clarity, there were balanced trees, like on in XGBoost, but symmetric oblivious trees used in CatBoost. Such trees are often not the best choice to learn on one-hot-encoded datasets, so exploring pruning strategy on this type of trees also requires an in-depth study of what type of trees and when we need to use for such data. It is interesting, but there is enough space and we will divert from the main topic — PolyTree framework introduction.

**Reviewer #3**

*Why the level of dependency grows with the number of conditions in M.* PolyTree decomposition is similar to $n$-way ANOVA decomposition, where dependence for factors $x, y, z$ is decomposed to main effects $(x, y, z)$ and their iterations $(xy, yz, xyz)$. In ANOVA $x, y, z$ are categorical factors, while in PolyTree it is right split indicator functions. *- Authors do not mentioned if the cross validation is applied or not. This latter should be applied.* The cross-validation is used during parameter tuning, but the final metric is computed on the fixed independent test set (experiment simulates practical usage of GBDT — choose hyper-parameters / model via cross-validation and then use it in production). *For section 3 relative to theoretical analysis, experiments remain poor, testing on only one set (Higg dataset) is insufficient to make reliable conclusions. More data sets should be tested.* This example shows, that there are certain type of problems, where to get the better model we need to use one type of tree shapes and are able to benefit from the other shape during exploitation. We don't insist that this situation is universal, but just show its existence. Systematic study what tree shape needed for what problem, and what benefits could be achieved for the specific problem at hand, requires space; we are working on this as follow-up work. *Why what is tested in Table 1 is only sets of binary classes. What about sets with multiple classes.* Its a topic for an individual study. There are several ways of how to perform multiclassification in boosting. For some types (like one-vs-all) the problem reduces to binary one, while for others not. There could be several ways to generalized PolyTree for multiclass, and each one should be explored. But this article introduces PolyTree framework and we don't have enough space to explore this problem in details. *In Tabe 1, How to explain that there is no significant improvement In AUC criterion between trained ensembles and pruned ensembles. In fact, you get almost the same AUC for both methods* For the trained ensemble, we have used the early-stopping strategy. GBDT with early stopping usually provided the state-of-the-art model. Thus, this model provides a very strong baseline that is hard to beat with the smaller model.

[Meta-Review · NeurIPS 2019]

This paper is borderline. Before rebuttal, one reviewer was very negative but then he changed his mind after the authors' response that clarified some of his misunderstandings. Now, all three reviewers are leaning towards acceptance. I thus follow them and recommend acceptance. I think the proposed transformation is novel and offers an interesting new way of interpreting tree forests, which can lead to new algorithmic solutions (e.g., for pruning and model interpretability as illustrated in the paper). The empirical evaluation is clearly weak, as highlighted by the reviewers, but I agree with the authors that the main contribution is the proposed transformation and that the experiments are merely provided as an illustration of the possibilities offered by this transformation. Given the reviews and my reading of the paper, I however ask the authors to make the following mandatory changes in the final version of their paper: - The authors should make all changes promised in their response and they should take into account all additional reviewers' suggestions to make the paper more clear. In addition, the different kinds of trees used (oblivious, symmetric, balanced, etc.) should be clearly defined as they are not so standard. - Section 5 should be clarified. I have the following questions: * It's not clear how the models are constructed. Do you build one model for each class (versus the others) or a single multi-class model? How exactly is a figure obtained for each class in Figures 3 and 4. * How do you apply Breiman's permutation based VI? This method originally estimates loss increase on out-of-bag samples when features are randomly permuted. How do you apply it on boosting models that do not use bootstrap sampling? Which loss do you use? I'm surprised that you get so many pixels with a negative importance value with this method in Figure 3. I expected this variable importance measure to be mostly positive for all pixels. * Accuracies of the models should be reported. Are we assessing features from a good enough model? * Not enough intuition is given for the metric in (8). Given that the averaged term can be negative or positive depending on the example x, couldn't v(k) be very small (or null) even when a feature is important? The authors should comment on that potential limitation in the paper. - I think that the authors should consider extending the pseudocode in Algorithm 1 (in the supplementary material) to make the subroutines more clear (SymmetricTree, IsSubset, and AddMonomialToTree). - The authors should change the name of their representation. The name 'polytree' will lead to confusion as 'polytree' is already used to refer to a specific graph structure, which has been used furthermore in machine learning papers (in the field of graphical models). - Overall, the paper needs to be carefully proofread as there are several typos in the text but also in formulas. - Citations are not properly formatted: they should be numbered in the NeurIPS style and less details must be given in the reference list (no urls, no ISSN, etc.). Be sure to use the correct format.